# Clinical Approaches for Mitochondrial Diseases

**DOI:** 10.3390/cells12202494

**Published:** 2023-10-20

**Authors:** Seongho Hong, Sanghun Kim, Kyoungmi Kim, Hyunji Lee

**Affiliations:** 1Korea Mouse Phenotyping Center, Seoul National University, Seoul 08826, Republic of Korea; anjgkwl0121@naver.com; 2Department of Medicine, Korea University College of Medicine, Seoul 02708, Republic of Korea; 3Laboratory Animal Resource and Research Center, Korea Research Institute of Bioscience and Biotechnology, Cheongju 28116, Republic of Korea; sanghun327@kribb.re.kr; 4College of Veterinary Medicine and Research Institute of Veterinary Medicine, Chungbuk National University, Cheongju 28644, Republic of Korea; 5Department of Biomedical Sciences, Korea University College of Medicine, Seoul 02841, Republic of Korea; 6Department of Physiology, Korea University College of Medicine, Seoul 02841, Republic of Korea

**Keywords:** mitochondrial diseases, mitochondrial therapy, clinical trials

## Abstract

Mitochondria are subcontractors dedicated to energy production within cells. In human mitochondria, almost all mitochondrial proteins originate from the nucleus, except for 13 subunit proteins that make up the crucial system required to perform ‘oxidative phosphorylation (OX PHOS)’, which are expressed by the mitochondria’s self-contained DNA. Mitochondrial DNA (mtDNA) also encodes 2 rRNA and 22 tRNA species. Mitochondrial DNA replicates almost autonomously, independent of the nucleus, and its heredity follows a non-Mendelian pattern, exclusively passing from mother to children. Numerous studies have identified mtDNA mutation-related genetic diseases. The consequences of various types of mtDNA mutations, including insertions, deletions, and single base-pair mutations, are studied to reveal their relationship to mitochondrial diseases. Most mitochondrial diseases exhibit fatal symptoms, leading to ongoing therapeutic research with diverse approaches such as stimulating the defective OXPHOS system, mitochondrial replacement, and allotropic expression of defective enzymes. This review provides detailed information on two topics: (1) mitochondrial diseases caused by mtDNA mutations, and (2) the mechanisms of current treatments for mitochondrial diseases and clinical trials.

## 1. Introduction

Mitochondrial disease refers to conditions caused by mitochondrial dysfunction [1]. Due to the distribution of mitochondria in every cell of the body, organ defects and symptoms vary widely. However, there is one clear commonality: the inheritance of mutated DNA that encodes major components of oxidative phosphorylation (OXPHOS), leading to a loss of mitochondrial function [2]. Mitochondria are organelles composed of numerous nucleus-encoded proteins with various roles [3,4,5], as well as 13 self-encoded proteins crucial for OXPHOS, along with 2 self-encoded rRNAs and 22 tRNAs [6]. Because all mitochondrially encoded proteins play important roles in mitochondria, mutations in coding genes within mitochondria can directly lead to mitochondrial dysfunction [7].

Unlike genetic disorders associated with nuclear genomes, the replication of mutated mtDNA does not exhibit patterns of Mendelian inheritance due to the independent fusion-fission activity of mitochondria. Mitochondrial DNA-related mitochondrial disease does not require the 100% homoplasmy of mutated DNA in an organism. Despite the heteroplasmic distribution being diverse across individuals and types of diseases, there is considered to be a certain threshold of mtDNA mutation load required to exhibit symptoms of diseases [8,9]. Correlations between heteroplasmic levels and the expression of symptoms have also been reported in case studies involving a large amount of screening of mitochondrial disease patients [10,11]. DNA sequencing methods have rapidly advanced, making it easier to find disease-inducing DNA mutations and even distinguish every single mtDNA sequence in a cell [12]. Yet, due to the difficulty of isolating individual mitochondria, there is a greater demand for studies to reveal the ambiguous dynamics of the mitochondrial processing of mtDNA. One thing that is not ambiguous is the notable prevalence of mitochondrial diseases [13,14,15]. Therefore, we are facing the need to solve this threatening problem affecting our populations.

Because mitochondrial diseases involve different types of dysfunctions in mitochondria, approaches for the treatment of mitochondrial diseases also vary. These approaches include replacing the cytoplasm containing defective mitochondria with healthy mitochondria-containing cytoplasm through oocyte spindle transfer [16], targeting the underlying cause of mitochondrial disease by converting pathogenic point-mutated mtDNA to normal mtDNA [17,18,19], the use of chemical compounds to stimulate electron transfer chain in OXPHOS by bypassing the malfunctioning complex [20,21], and even researching phenolic compounds in our diet that have shown potential for reducing ROS through their antioxidant activity [22,23,24,25].

Mitochondrial defect also affects the closely associated protein, AMPK [26]. This, in turn, impacts the AMPK’s protein activity of regulating antitumor immunity [27,28]. In terms of this, research is being conducted to induce OXPHOS depression in mitochondria, which affects AMPK and further promotes cancer immunotherapy [29,30]. Studies to control the population of dysfunctional mitochondria by utilizing the involvement of AMPK protein in mitochondrial biogenesis are also in progress [31,32]. In the context of mitochondria-related autophagic processes, which involve the regulation of dysfunctional mitochondria populations, this approach holds the potential for treating mitochondrial diseases and is actively being investigated [33,34,35,36,37].

Countless studies are being conducted in the field of mitochondrial disease, but only a few studies have been completed with considerable efficiency in terms of restoring defective mitochondria, and an even more limited portion of groups have reached clinical trials with their treatments. This review focuses on treatments that are currently undergoing clinical trials for mitochondrial disease, with initial explanations of several mitochondrial diseases.

## 2. Features of Mitochondrial Disease

One notable aspect of mitochondrial disease is that, despite its diverse causes at the cellular level, symptoms are generally expressed through mitochondrial dysfunction. Consequently, although the main symptoms of mitochondrial disease vary depending on the type of disease, they appear to share common symptoms in the broader category of encephalomyopathy. Therefore, it is important to identify the defective mechanisms at the cellular level in order to understand the causes of mitochondrial disease.

Mitochondria are subcellular organelles known for producing ATP, which is used for cellular energy, through OXPHOS. Mitochondrial disease occurs when there are defects in the proteins involved in OXPHOS or other proteins related to mitochondrial function. Researchers have shown a correlation between mitochondrial performance and high-energy demanding cells, and mitochondrial proteome also varies depending on the tissue [38,39,40,41,42,43,44,45,46]. Therefore, mitochondrial dysfunction can be regarded as a deficiency in cellular energy, which leads to energy deficiency in nerve cells or myocytes, appearing as the main cause of mitochondrial disease. However, studies on patients with mitochondrial disease have reported that the main cause is often associated with lactic acidosis. Dysfunction in the electron transport chain leads to a decrease in ATP production, and low ATP level further increase glycolysis, resulting in an overproduction of pyruvate, which can be further reduced and converted to lactate [47,48,49].

Mitochondrial proteins can be encoded by both nuclear DNA and mtDNA, thus mitochondrial disease can also result from mutations in either or both types of DNA. The inheritance of mitochondrial diseases caused by nuclear DNA mutation can be easily identified by examining family histories, as the symptoms follow the rules of Mendelian inheritance. However, mitochondria have their own autonomous process of duplication and DNA replication. Therefore, mutated mtDNA can exist in a state of heteroplasmy, leading to cellular dysfunction and an increase in the expression of mitochondrial disease symptoms. The manifestation of these symptoms is expected to depend on a certain threshold of heteroplasmy, which varies depending on the type of disease and the individual carrying the mutated mtDNA [50,51,52]. Case studies have shown that the amount of mutated mtDNA can vary among patients with the same symptoms, and higher levels of heteroplasmy do not always correlate with the severity of symptoms. Even in the case of monozygotic twins with mitochondrial disease, different symptoms can be observed, suggesting that there are additional factors beyond DNA mutations that contribute to the expression of symptoms [53].

In case studies involving large amounts of patient data, a convincing correlation was found between heteroplasmy level and the age of onset [10,11], but no correlation was observed between symptoms. Due to the elusive nature of mitochondrial pathologies, it is important to understand the mechanisms and causes that trigger them.

## 3. Types of Mitochondrial Disease

Since mitochondrial diseases tend to have numerous causes beyond their prominent cause, this paper only focused on major types of disease-causing mutations. The description of mitochondrial disease with notable symptoms and case studies is given below (Table 1, Figure 1).

Leber hereditary optic neuropathy (LHON) is known to exhibit eye-related symptoms, such as sequential vision loss and dyschromatopsia. Patients with LHON also have a higher likelihood of experiencing neurological abnormalities, including peripheral neuropathy and myopathy, similar to other mitochondrial diseases, compared to the general population [80,81,82]. However, these symptoms do not manifest as a chain of events. Consequently, LHON is more commonly diagnosed based on optic neuropathy-related symptoms.

In LHON, up to 90% of clinical cases are associated with three mutations: m.G11778A (MT-ND4), m.A3460G (MT-ND1), and m.T14484C (MT-ND6). Among these mutations, m.G11778A is the most dominant [83]. ND1–6 are the components of the NADH-ubiquinone oxidoreductase complex (complex 1), and therefore, mutations associated with LHON cause defects in complex 1, which has the function of catalyzing reactions in the electron transport chain.

Patients with LHON have high levels of mutated DNA heteroplasmy, almost reaching the level of homoplasmy [83,84]. Also unaffected carriers of LHON-inducing mtDNA mutations can be easily found in the families with LHON patients [85,86,87,88], from which could be further inferred that LHON-inducing mtDNA mutations do not seem to be critical for the functioning of mitochondria. Therefore, several studies have been conducted to discover the triggers of symptom expression [89,90,91,92,93,94], but the underlying factors of LHON remain elusive.

The LHON mutation-derived mitochondrial defects mostly result in optic neuropathy, affecting optic nerve, axons, retinal ganglion cells, and photoreceptor cells [83,95]. Case studies have revealed affected retinal nerve fiber layers and impaired photoreceptor functions. Therefore, ocular symptoms are experienced, such as sequential loss of visual acuity and dyschromatopsia [54,55,56].

Myoclonic epilepsy with ragged red fiber (MERRF) is a multi-system disease characterized by progressive myoclonus and seizures [96]. One of the most common mutation among the patients with different types of mtDNA mutations is m.A8344G, which affects the mitochondrial tRNA lysine [97,98,99,100,101]. Another prominent mutation in the same gene is m.G8363A, along with the m.A3243G, m.G3255A, and m.T3291C mutations, which have also been reported to cause MERRF with a defects in tRNA leucine [57,58,59,99,102,103]. Mutations in tRNA-coding genes induce the global impairment of mtDNA-encoded proteins rather than affecting certain complexes or pathways [101,104,105]. This can result in the overall dysfunction of mitochondria. In case studies involving MERRF patients with tRNA lysine mutations, various symptoms were reported, such as unsteadiness of gait, optic atrophy, slurred speech, sensorineural hearing loss, and limb weakness [57,58,59].

One well-known mitochondrial disease is mitochondrial encephalomyopathy with lactic acidosis and stroke-like episodes (MELAS). It is also caused by a mutation in the mitochondria-encoded tRNA leucine, specifically the m.A3243G mutation, which leads to mitochondrial dysfunction and similar effects as MERRF, affecting the synthesis of all mtDNA-encoded proteins [106,107,108,109].

MELAS is a multi-systemic disease; therefore, misdiagnosis was common until the utilization of DNA analysis on relatives or families of the patients. In one case study, a patient was initially diagnosed with diabetes at an early age, but the cause of the gradual development of diabetes remained elusive until the patient’s sister was diagnosed with maternally inherited diabetes and deafness (MIDD) [60]. Subsequent medical studies were then conducted, resulting in the diagnosis of MELAS. In total, it took 25 years from the initial diagnosis of diabetes for the patient to be diagnosed with MELAS. Stroke-like episodes are one of the main aspects used to diagnose MELAS [60,61,110], and symptoms of MELAS are commonly considered to begin before the age of 40 [111]. Recent studies have revealed that the disease can develop after the age of 40 in a small number of patients [112], and adult-onset cases of MELAS have also been reported [62,113,114,115,116]. In cases of late onset of the disease, most patients have low levels of heteroplasmy or significant variations in various tissues. Similar to other mitochondrial diseases, the improvement of symptoms is rare, and in most cases, the symptoms deteriorate rapidly.

Maternally inherited diabetes and deafness (MIDD) is caused by the same mutation as MELAS (m.A3243G). Since both diseases show a broad spectrum of symptoms, it is hard to differentiate between them. However, MIDD is characterized by sensorineural hearing loss and decreased insulin secretion [117,118,119]. A case study conducted with 161 patients with MIDD revealed a correlation between the age of onset of diabetes and the heteroplasmy level of the mutated mtDNA (m.A3243G).

There remains a question regarding the different consequences of MELAS and MIDD despite being associated with the same mutation. Previous case studies have shown that MIDD patients are likely to exhibit a low heteroplasmy of mutated DNA [120,121,122]; however, apart from this finding, no other direct evidence has been found to provide a clear explanation.

Leigh syndrome seems to be caused by a lack of energy in nerve cells [123], resulting in the failure of neuronal morphogenesis and maturation [124,125]. Numerous pathogenic mutations, more than 75 in mtDNA and nuclear DNA, have been reported [126,127]. Of the various causes, a characteristic mutation in mtDNA that is typically observed in Leigh syndrome is the mutation in the MT-ATP6 coding gene, and numerous types of mutations have been identified as pathogenic [128,129]. The defect in MT-ATP6 leads to the dysfunction of mitochondrial complex 5, which converts ADP to ATP, eventually resulting in a lack of energy.

Typical symptoms of Leigh syndrome include motor development delay, muscle disorder, and cardiac dysfunction. Case studies have shown that Leigh syndrome also exhibits a wide spectrum of mitochondrial disease-associated symptoms [130]. Although one of the main symptoms of Leigh syndrome is delayed development, some case studies have reported patients with a late onset of the first symptom [74,131,132].

Pearson syndrome, Kearns–Sayre syndrome (KSS), and chronic progressive external ophthalmoplegia (CPEO) are mtDNA deletion-induced mitochondrial diseases. Unlike other mitochondrial diseases, which show an increased distribution of mutated mtDNA in offspring, mtDNA deletion-related diseases appear to have less dependency on maternal inheritance [133,134,135]. Yet, the dynamics of mtDNA deletions are not fully understood, and there are potential hypotheses and supportive studies on the mechanisms [136,137].

The positions of mtDNA deletions are random, and the deletions are usually large, leading to overall mitochondrial dysfunction [133,138,139]. Three conditions share similar phenotypes, but in the case of Pearson’s syndrome, severe symptoms usually begin in infancy and causing death before the age of four [140]. Children who survive develop Kearns–Sayre syndrome (KSS) eventually experience a reduced quality of life and sudden death [141]. KSS and CPEO can be referred to as KSS minus or CPEO plus based on the severity of symptoms [142]. KSS also exhibits symptoms of progressive external ophthalmoplegia (PEO), but if a patient expresses isolated symptoms of PEO, they are referred to as CPEO [142]. Ultimately, since both diseases are associated with the same mtDNA deletion, a patient diagnosed with CPEO could experience a spectrum of symptoms in a multi-systemic disease and therefore become a patient with KSS [143,144].

Previously described mitochondrial diseases are mainly caused by mutations in mtDNA. There are also diseases that can result from both nuclear and mitochondrial mutations, such as Leigh syndrome and CPEO. Barth syndrome and Friedreich’s ataxia are mitochondrial diseases caused by mutations in nuclear DNA. Mutations in both cases do not disrupt the mitochondrial electron transport chain complexes, as they do not affect the composing subunits. Instead, these mutations affect proteins regulating the overall dynamics of mitochondria. Hence, they also result in the same dysfunctions observed in other mitochondrial diseases.

Due to the lack of treatments for mitochondrial diseases, the clinical approach in case studies typically involves a combination of known antioxidants, coenzyme Q10, and vitamins. Hence, there have been no significant improvements in the severe symptoms, except in the specific cases of LHON, which predominantly exhibits symptoms related to optic neuropathy, and other mitochondrial diseases exhibiting similar symptoms in the spectrum of encephalomyopathy. Therefore, a single treatment can potentially be applicable to various diseases if the restoration of disordered mitochondria can be demonstrated.

## 4. Clinical Approach with Treatments Involving Chemical Compounds

Intracellular interactions of disease-targeting molecules often affect various cellular mechanisms in addition to their primary target pathway. As a result, diverse clinical approaches can be conducted using the same treatment. This review specifically focuses on treatments aimed at rescuing dysfunctional mitochondria that are currently undergoing trials (Table 2, Figure 2).

Until now, the most widely used chemical compound in diseases caused by mitochondrial dysfunction has been coenzyme Q10. Coenzyme Q10 is a self-generated resource and also an FDA-approved dietary supplement [177,178], which is well-known for its role as a powerful antioxidant in cells [146,147]. Self-generated coenzyme Q10 serves as a diffusible electron carrier in the mitochondria. Based on the known functions of coenzyme Q10, various research studies have analyzed and reported the potential of coenzyme Q10 for therapeutic application in mitochondrial diseases [179,180]. A phase 3 clinical trial for mitochondrial disease was conducted and completed in 2013 (NCT number, NCT00432744). However, coenzyme Q10 still lacks evidence of significant results in restoring mitochondrial dysfunction, and its therapeutic effect remains elusive. In a phase 3 clinical trial for Parkinson’s disease, which can also be regarded as a pathology related to mitochondrial dysfunction [181], no beneficial evidence was found for rescuing defective mitochondria. Due to its limited success in clinical approaches for diseases, coenzyme Q10 is not yet FDA-approved for medical treatment of disease [178,182,183]. Although coenzyme Q10 (CoQ10) cannot be used to treat any medical condition, there is no issue regarding its use as a nutrient supplement. Therefore, it is often used as part of a cocktail of nutritional supplements, mixed with other vitamins for mitochondrial diseases.

Based on the partially beneficial effects of coenzyme Q10, related chemical compounds are being developed for treatments. Idebenone is the most recognized drug for mitochondrial diseases. Idebenone is a short-chain hydrosoluble quinone [184], which overcomes the disadvantage of insoluble coenzyme Q10 and has demonstrated strong antioxidant activity. It also functions as an electron carrier in the mitochondrial electron transfer [148,149,150,151,183]. Further studies have found that Idebenone can overcome complex 1 deficiency in patients with LHON by directly transferring electrons to the complex 3, bypassing complex 1. This restores the function of the electron transfer chain and normalizes the production of cellular energy [148,149,150,151]. Considering that almost all mitochondrial diseases have defects in complex 1 or the overall electron transfer chain, Idebenone could be used for the direct treatment of complex 1 defect, and it also shows promise for treating electron transport chain dysfunction. Clinical studies for various mitochondrial diseases are currently ongoing [185]. For the medical treatment of LHON patients, a phase 4 clinical trial has been completed (NCT02774005) and Raxone (Idebenone) has been approved by the European Medicines Agency (EMA) for use with LHON patients (product number for EMA, EMEA/H/C/003834). However, it has not received approval from the FDA.

Vatiquinone is a quinone-based molecule that shares structural similarities with coenzyme Q10 and Idebenone but exhibits a higher level of protectant activity against oxidative stress [186]. Further studies have revealed that Vatiquinone modulates the inflammation process and depletes glutathione (GSH) at the cellular level by acting as an inhibitor of 15-lipoxygenase (15-LO) [152]. The reduction in ROS activity has also been demonstrated in human samples from patients with Leigh syndrome and healthy subjects. Vatiquinone increased GSH levels and decreased oxidized GSH levels, leading to a significant reduction in ROS levels [153,154,187].

With promising results in the performance of defected mitochondria, clinical trials have progressed. Several supporting results were found in a trial of patients considered to be within 90 days of end-of-life [188]. Visual improvement was confirmed in a young patient with Leigh syndrome [189]. Furthermore, a phase 2/3 clinical trial for mitochondrial disease with refractory epilepsy was conducted (NCT04378075), but it was announced to have failed to achieve its primary endpoint of reducing observable motor seizures. However, phase 3 clinical trials for the safety study of other symptoms are currently ongoing. There is a trial underway with Vatiquinone-exposed patients for the treatment of Friedreich ataxia (NCT05515536) and another trial for the broad aspects of mitochondrial diseases (NCT05218655).

NAD+ is an important factor in the energy production by the mitochondria. The study of NAD+ precursors has shown potential in enhancing mitochondrial functions. In addition, a decrease in NAD+ levels is observed in mitochondrial diseases, which is caused by deficiencies in the mitochondrial electron transfer chain. This decrease in NAD+ causes a disruption in redox homeostasis and energy metabolism, ultimately affecting various signaling pathways in cells [156,190]. Various studies have shown that an increase in intracellular NAD+ levels can lead to an improvement in mitochondrial number and oxidative capacities [156]. Nicotinamide riboside (NR), a precursor of NAD+, has been studied for its potential to enhance mitochondrial function. It was expected to improve adiposity but failed to do so [191,192]. However, a few short-term follow-up studies reported the possibility of NR supplementation leading to physical performance improvements and the alleviation of minor symptoms related to circulatory system [191,192]. NR research in mice has focused on its effects on mitochondria, showing that it increases mitochondrial function, extends the lifespan of mice, and helps retain neuropathy [155,157,158]. In a human trial, the number of mitochondria increased, and the broad upregulation of NAD+ metabolism resulted in various enhancements of disease symptoms [193]. Currently, a phase 2 clinical trial is ongoing for patients with mitochondrial myopathy disorders.

KL1333 is a compound that acts as a NAD+ precursor through interaction with NAD(P)H quinone oxidoreductase 1 (NQO1), which is a NADH-to-NAD+ conversion enzyme [159]. It increases NAD+ levels and leads to enhancements. Currently, a phase 2 clinical study for mitochondrial disease is in progress (NCT05650229).

Peroxisome proliferator-activated receptors (PPARs) are transcription factors located in the nucleus that modulate several pathways through gene expression. PPARδ is a subtype of PPARs that is known to be highly expressed in skeletal muscle cells compared to other PPAR subtypes. It has a preference for increasing fatty acid oxidation and enhancing mitochondrial biogenesis [161,162,163,164,194]. Thus, in cases of mitochondrial disease where symptoms are observed in skeletal muscle, the use of PPARδ agonists could be a therapeutic option, which may increase bioenergetics in skeletal muscle. For the primary mitochondrial myopathy, which exhibits mitochondrial dysfunction-related symptoms especially in skeletal muscles, bocidelpar (ASP0367, clinical trial phase 2/3, NCT04641962) and mavodelpar (REN001, clinical trial phase 2, NCT04535609) are currently undergoing clinical trials as PPARδ agonists.

Elamipretide (SBT-272) has a distinct activity in restoring defective mitochondria compared to other chemical compounds described previously. The target of Elamipretide is the cardiolipin of the mitochondrial inner membrane [165], which has a role in maintaining the architecture and morphology of the mitochondria [166]. The proper structure of mitochondria is important for the proper functioning of mitochondrial proteins. Therefore, a defect in cardiolipin could lead to the impairment of mitochondrial structure, resulting in neural diseases [195,196,197]. Few studies have reported evidence that the recovery of cardiolipin levels could improve neuropathy [195,197], and more beneficial effects for mitochondrial disorders and heart failure have been observed [198,199,200].

Barth syndrome is a mitochondrial disease known to be caused by a defect in the Tafazzin protein, which further leads to cardiolipin abnormality [201]. In a preclinical study of Elamipretide, the network of mitochondria cristae was remedied, resulting in the improvement of bioenergetics dysfunction in rats [202]. The phase 3 clinical trial of Elamipretide in Barth syndrome (NCT03098797) has been completed, and another approach for recovering mitochondrial dysfunction is currently underway for primary mitochondrial myopathy, which is now in phase 3 of clinical trials (NCT05162768).

The disorder of cells caused by an increase in ROS is a direct result of mitochondrial defects. Sonlicromanol (KH176) is a compound that reduces ROS levels. It is a derivative of Trolox (water soluble Vitamin E) and possesses enhanced antioxidant properties [203,204]. Sonlicromanol mediates redox biology through its interaction with one of the major cellular signaling systems involved in antioxidant and redox processes, namely the thioredoxin system [160].

The successful rescue of a neural network within a high heteroplasmy load has been shown using patient-induced stem cells by modulating the neuronal transcriptome [205]. A clinical trial was conducted to rescue MELAS in a patient with the m.A3243G mutation (NCT04165239), and phase 2 of the trial has been completed.

Omaveloxolone (SKYCLARYS) is the only treatment with FDA approval for use in patients with mitochondrial disease (Friedreich’s ataxia). Friedreich’s ataxia is a disease characterized by defective Frataxin expression caused by a pathogenic repeat in the Frataxin coding gene, resulting in iron accumulation in mitochondria and increased oxidative stress in cells [167,168]. Omaveloxolone, which is a synthetic derivative of oleanolic acid, binds to Kelch-like ECH-associated protein 1 (Keap1), preventing the degradation of nuclear factor erythroid-2-related factor 2 (Nrf2). Nrf2 is a transcription factor that mediates antioxidant gene expression, represses pro-inflammatory gene expression, and enhances mitochondrial biogenesis. Thus, the application of Omaveloxolone restores cellular defects by protecting Nrf2. Omaveloxolone has shown a restoring effect on the disorder in electron transfer chain complex 1, rescuing mitochondrial functions [206,207]

Omaveloxolone is currently being used to treat patients with Friedreich’s ataxia. The FDA’s approval of Omaveloxolone may seem to indicate a greater pharmacological efficacy compared to other drugs currently under development for mitochondrial diseases. Even Raxone (Idebenone), which has demonstrated therapeutic effects for LHON disease in numerous case studies, has not yet received FDA approval. While Omaveloxolone is the first FDA-approved treatment for mitochondrial disease, there are conflicting opinions on its clinical effects, and the strength of the supporting efficacy data is still debatable [208].

Due to the approaches of drug treatments for mitochondrial diseases, most of the clinical effects occur at the level of defected pathways and do not affect DNA mutations, which are the original cause. Therefore, efficacy can vary among individuals, and the application of treatment needs to be continued throughout a lifetime.

## 5. Clinical Approach Using Non-Chemical Treatments

In terms of non-chemical treatments for mitochondrial diseases, there is a tendency to address the pathology by transferring healthy mitochondria or normal mitochondrial gene into cells with dysfunctional mitochondria, rather than targeting mitochondrial pathways (Table 3, Figure 2).

In the field of gene therapy, the adeno-associated virus (AAV) is known as a vehicle that can transduce packaged DNA (~4.7 kb) and enable allotropic expression in organisms [170,209,210,211].

A clinical approach for LHON patients, who have dysfunctional MT-ND4 in complex 1, involved a pre-clinical study using allotropic expression of normal MT-ND4 through mRNA delivery. A rodent model of LHON showed enhanced visual deficits, and the rescue of ATP synthesis was observed in patient-derived fibroblasts [170,171,172]. Subsequently, gene delivery using AAV was performed to achieve similar results through allotropic expression, resulting in certain improvements in a mouse model of LHON [171,212].

In the history of LUMEVOQ for LHON (rAAV2/2-ND4, GS010), a preclinical study was conducted using recombinant AAV serotype 2, which is commonly employed for transducing to retinal ganglion cell nuclei and does not cause retinal injury [172]. Application in human patients was carried out through the intravitreal injection (IVT) of human wild-type ND4-packaged rAAV2. Currently, a phase 3 clinical trial is underway for LHON patients with the m.G11778A mutation (NCT03293524), and some of the preliminary results are already providing evidence of improved visual acuity [213]. Additionally, evidence of contralateral improvement was found in unilateral treatment. Accordingly, viral gene transfer approaches are under research [214]. A group, using the same method of treatment for LHON, is currently in their clinical trial phase 2/3 (NR082, NCT03153293). Another group, using a slightly different approach with self-complementary AAV for packaging the vehicle [173], is currently conducting a phase 1 clinical trial (ScAAV2-P1ND4v2, NCT02161380).

Mitochondrial augmentation (MAT) is a therapeutic method that involves transferring normal exogenous mitochondria into cells with disordered mitochondria. Mitochondrial transfer between cells to restore respiration capability has been demonstrated by using mtDNA-mutated cells (A549) with hMSC or skin fibroblasts [215]. Further research found more evidence of mitochondrial transfer [216,217,218]. In addition, experiments have successfully restored mitochondrial defects via the ex vivo augmentation of patient-derived hematopoietic stem and progenitor cells (HSPCs) with normal donor exogenous mitochondria [174]. MAT treatment has also been conducted in human patients using patient-derived autologous CD34+ hematopoietic cells reinserted through intravenous infusion. Based on previous MAT treatments, a decrease in mtDNA heteroplasmy level was observed in the peripheral blood of patients (4/6), and some patients showed improvement in some parameters of physical examination [219].

MAT is a therapy that replaces disordered mitochondria in cells with normal mitochondria and is not limited to correcting specific pathogenic mutations. MAT (NCT03384420) for Pearson syndrome, which is caused by large and random mtDNA deletions, is currently undergoing phase 1/2 clinical trials. 

Another therapy option for the direct replacement of defective mitochondria is the use of patient-autologous mesoderm-derived stem cells, known as mesoangioblasts (MAB).

In a previous study, it was observed that MAB has low heteroplasmy loads despite having high loads of mutated mtDNA in skeletal muscle cells [176]. Mutated mtDNA seems to show a tendency to decrease in cultured satellite cells [175,220,221]. It has the advantage that when it is intra-arterially administrated to patients systemically and fused with damaged muscles, it contributes to inducing muscle regeneration due to its high myogenic potential [222,223]. In the first phase of the clinical trial, treatment with autologous MAB was conducted accompanied by ex vivo cultured patient-derived MAB. Currently, it is in a phase 2 clinical trial for the treatment of mitochondrial myopathy in patients with the m.A3243G mutation (NCT05962333).

For the treatment of unborn children whose parents have mitochondrial disease, mitochondrial replacement techniques (MRT, not listed in the table) are promising procedures, which can be used to replace almost all disordered mitochondria with healthy ones in human oocytes or zygotes [224,225]. Except for the fertilization process, various types of MRTs involve replacing the maternal nuclear DNA with the nuclear DNA in the donor’s cells. Despite the successful results of MRT operations [16], there have been debatable outcomes, such as the reversion of mutated DNA [226,227] and other ethical concerns [228]. Ultimately, the FDA has announced that performing MRT is not permitted in the USA.

Although clinical approaches using non-chemical treatments demonstrate their effect through numerous case studies, restoring efficiency remains elusive. Nevertheless, their approach towards achieving permanent effects, as opposed to chemical treatments, holds promise for a potential cure for mitochondrial diseases.

## 6. Targeted Genome Editing

In the field of clinical approaches for genetic diseases caused by nuclear DNA, there are exceptional genetic editing tools with CRISPR-Cas-based technology, which can target any sequence using target-complementary guide RNAs. In research on CRISPR-Cas-based base editors, which can edit single pathogenic base pairs with Cas protein and connected deaminase proteins [229], clinical trials are currently underway (NCT05456880, clinical study phase 1/2; NCT05885464, also phase 1/2).

In the case of mutations in mtDNA, the CRISPR-Cas system appears to be ineffective due to the mitochondrial import dynamics that prevent the transport of the guide RNA [230]. As a result, the current methods for targeted base editing of mutated mtDNA rely on approaches used prior to the development of the CRISPR-Cas system.

The transcription activator-like effector nuclease (TALEN) uses an array of TALE proteins to target specific sequences. Each TALE protein can recognize three nucleotides based on its RVD motifs [231,232,233]. Since TALE is formed by proteins, it can be transported to mitochondria using mitochondrial targeting signal amino acid sequences. Previous methods focused on TALEN for cutting DNA. Base editing with TALE array was not conducted before the emergence of CRISPR-Cas-related base editors (Figure 3).

The first application of base editing in mitochondria was conducted using a combination of bacterial cytidine deaminase toxin and TALE array [17]. In the progress of producing a mitochondrial cytosine base editor, the DddA toxin was divided in half to eliminate toxicity and become active after recruitment at the target site. Additionally, an uracil glycosylase inhibitor (UGI) was used to inhibit the repair mechanism of cytosine deamination. These modified DddA toxins are referred to as DddA-derived cytosine base editors (DdCBEs). After successfully applying targeted base conversion in mitochondria using DdCBEs, an in vivo mouse model with an induced pathogenic mutation of mtDNA was generated [234].

Although it was expected to be designed for the C-to-T base editor, the preference for base conversion only occurs at the 5′ TC motif. To overcome this bias, an improved editor called DddA11 with an expanded target range was developed [235]. Further development was conducted to enhance efficiency using a nuclear export signal, taking advantage of its preference to localize in the mitochondria rather than the nucleus [236].

The mitochondrial base editor is commonly used to induce pathogenic mutations in rodents to establish a mouse model of mitochondrial disease with disease phenotypes [234,236,237,238]. The use of DdCBEs in an in vivo model is usually conducted through microinjection at the zygote stage but cannot effectively target all mitochondria, resulting in a heteroplasmic distribution of mutated mtDNA. Since a high mutation load is required for the expression of mitochondrial diseases, there is a possibility of creating a mouse model with higher heteroplasmy levels through the continuous breeding of mutant mice [239]. In terms of effectiveness, improvements should focus on enhancing the overall impact on intracellular mtDNA.

After the demonstration of target-specific C-to-T base editing in mitochondria, an A-to-G base editor was also engineered using bacterial tRNA adenosine deaminase (TadA), which has shown its ability to mediate A-to-G conversion with the CRISPR-Cas system [240,241]. The first mitochondrial TALE-based A-to-G base editor, TALE-linked deaminases (TALED), uses an improved version of TadA, called TadA8e for base editing [18,242]. Since TadA lacks the ability to create a DNA bubble needed for its activity, TALED uses DddAtox splits to prepare single strands for TadA deamination [18]. Toxicity was the reason for splitting DddAtox in half. A monomeric form of TALED was also produced by inactivating the full DddAtox, but this change did not increase base editing efficiency.

One of the latest tools, mitochondrial DNA base editors (mitoBEs) uses mitochondria-derived nickase protein to create a suitable DNA structure for deaminase activity [19]. Both C-to-T and A-to-G conversions have been demonstrated using the deaminase protein rAPOBEC1 (for C-to-T) [229] and the TadA8e-V106W variant.

Given that researchers are discovering various editors using the CRISPR-Cas system [243,244,245], the development of a base editor based on CRISPR-Cas for mtDNA seems imminent. Improvements in mitochondrial DNA editing systems may lead to promising treatments and a permanent cure for mitochondrial diseases. However, many obstacles remain on the path to developing mitochondrial base editors for human clinical treatment. Diverse types of mitochondrial base editors have shown inconsistent performances with target-dependent efficiencies [17,18,19,235], as well as bystander editing within the target window. Genome-wide off-target effects have been observed in human cells [246] and mouse embryos [247], which is a major concern as it could lead to unexpected disorders. In addition, without the cytosine base editor DdCBE, other base editors still lack studies on in vivo models for further research. Nevertheless, mitochondrial base editing is still expected to be an attractive treatment method as it can correct mutations that cause mitochondrial disease and restore normal function.

## 7. Conclusions

Patients diagnosed with mitochondrial diseases are extremely rare, making it difficult to conduct sufficient case studies. Additionally, individuals with mitochondrial disease exhibit fatal symptoms, which poses a challenge for researchers attempting to study them. Therefore, it is a challenge to overcome these obstacles and develop effective treatments for mitochondrial disease.

In the past, it was impossible to recognize the underlying causes of mitochondrial disease. As a result, the approaches used to resolve it were also ambiguous, such as advising mtDNA mutation-harboring patients to exercise more to regenerate muscles, without considering their intolerance to physical activities [248,249]. While symptoms with a small portion of mutant mtDNA or age-related gradual mitochondrial defection can be managed with more nutritious diets or exercise, these methods cannot replace medical treatments.

Today, there are no obstacles in identifying DNA mutations in patients to find the underlying causes of mitochondrial diseases. Research on the dynamics of mitochondria continues, which leads us to aim for more specific pathways. Pharmacological advances are also leading to the discovery of new treatments for rare diseases.

There are several ongoing research studies focused on rescuing dysfunctional mitochondria that are not fully described in this review. For example, restoring mtDNA depletion by treating with deoxypyrimidine (deoxycytidine and deoxythymidine) to bypass the pyrimidine salvage pathway of mitochondria [169,250] is one clinical approach being used in human trials (NCT04802707, phase 2). OMT-28, a mitochondrial rescue agent involving Omega-3 fatty acid-related activity, is also under investigation (NCT05972954, Phase 2).

Although various drugs are currently undergoing human trials, many of them still seem to be ambiguous in terms of their effects on mitochondrial diseases, due to their indirect activities on the main causes. Mitochondrial diseases are inherited and permanent, and the known drugs for mitochondrial diseases provide only temporary relief rather than a cure. Therefore, a further approach to treatment should focus on addressing the underlying causes permanently.

MAT, or treatments with autologous MABs, could be regarded as one of the approaches that are the closest to human treatments involving the permanent replacement of defected mitochondria. While these therapies demonstrated notable effects in restoring defective mitochondria-harboring cells, no sufficient changes in the mitochondrial population that could completely cure mitochondrial diseases have been reported. AAV-mediated gene therapy could also be a potential approach for the treatment of mitochondrial diseases. Since spinal muscular atrophy, a genetic disease caused by nuclear DNA, has an FDA-approved AAV-mediated treatment, Zolgensma (Onasemnogene abeparvovec; FDA submission tracking number: 125694), and developing mitochondrial gene therapies using AAV seems to be promising approach. Many of these types of non-chemical treatments have been developed; however, they still have limitations in that their approach is more targeted towards specific tissues with direct symptoms rather than the entire body’s mitochondria. Mutated mitochondria are typically spread throughout the entire body in an individual, rather than being tissue-specific. Therefore, even if mitochondrial populations in one tissue are restored, it might lead to subsequent disorders in other tissues. Considering this, further research is required.

Expressed symptoms of mitochondrial diseases usually result in fatal conditions. However, current treatments for mitochondrial diseases are still considered remarkable if they show any effects on disease or improve some parameters. Despite the rarity of patients expressing severe symptoms, disease-inducing pathogenic mutations can be inherited without symptoms by many individuals. Without analyzing mtDNA sequences, there is a possibility that anyone could also be an unaffected carrier of mitochondrial diseases. Therefore, more attention is needed in the research of mitochondrial diseases and the clinical approaches to them.

## Figures and Tables

**Figure 1 cells-12-02494-f001:**
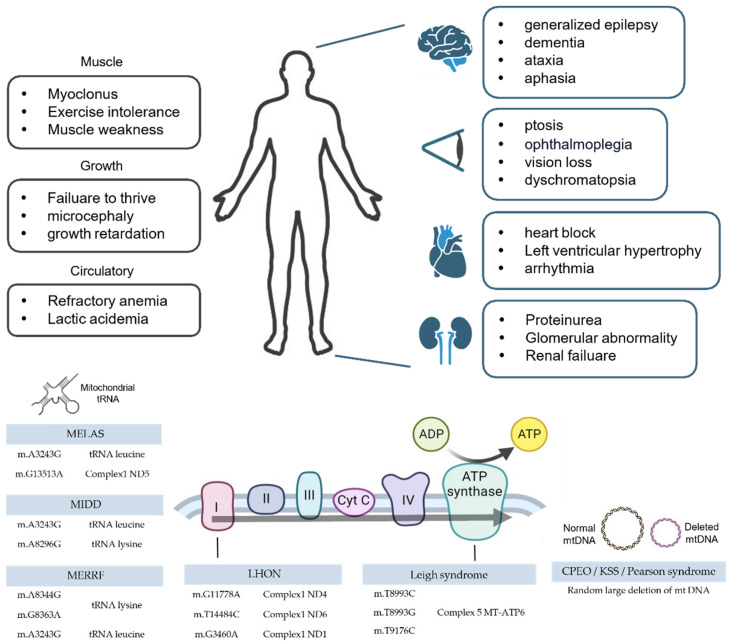
Symptoms and causes of mitochondrial diseases. The upper side of the figure displays commonly diagnosed symptoms, with symptoms affecting the entire body placed on the left and organ-specific symptoms on the right. As mitochondrial diseases have various causes, only well-known pathogenic mtDNA mutations are listed. A line and brief illustration indicate a correlation between pathogenic mutations and the affected sites. Details can be found in Table 1, Section 3.

**Figure 2 cells-12-02494-f002:**
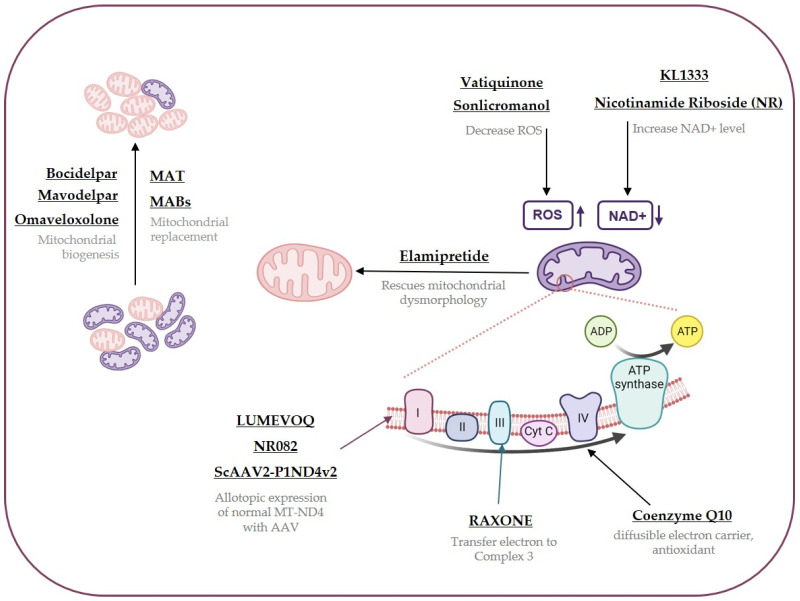
Interactions of treatments for mitochondrial diseases. Targeting points of drugs are indicated with arrows with brief illustrations. The treatments are displayed using their known names and features, and further detailed mechanisms for each treatment can be found in Section 4 and Section 5 and Table 2 and Table 3.

**Figure 3 cells-12-02494-f003:**
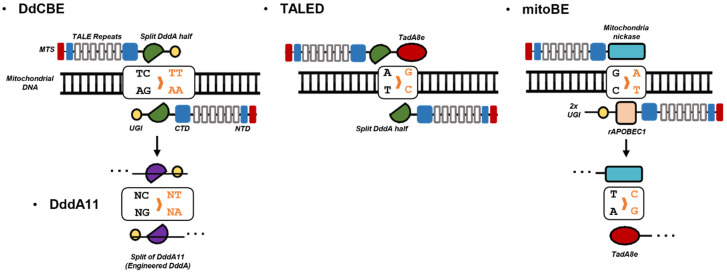
Currently developed mitochondrial base editor. Identical proteins are displayed with the same shape and color. Target and edited nucleotides are marked using boxes, with the edited nucleotides shown in orange. An arrow indicates another form of the same base editor with different deaminase activity. Further details can be found in Section 6.

**Table 1 cells-12-02494-t001:** Known diseases with mitochondrial malfunction.

Disease	Main Symptoms	Case Reports References	Cause
LHON (Leber hereditary optic neuropathy)	Vision loss (retinal ganglion cells and axon loss)	Significant decrease in visual acuity, dyschromatopsia [54,55,56]	m.T14484C (Complex1 ND6)
			m.G3460A (Complex1 ND1)
			m.G11778A (Complex1 ND4)
MERRF (Myoclonic epilepsy with ragged red fiber)	Myoclonus, generalized epilepsy, ataxia, myopathy, exercise intolerance	Bilateral primary optic atrophy, sensorineural hearing loss, gait unsteadiness [57]	m.A8344G (tRNA lysine)
		Slurred speech, fatigue, hearing loss, limb weakness [58]	m.G8363A (tRNA lysine)
		Dysphonia, memory loss, stroke-like episode [59]	m.A3243G (tRNA leucine)
MELAS (Mitochondrial encephalomyopathy with lactic acidosis and stroke-like episodes)	Muscle weakness in whole body, dementia, aphasia, myoclonus, ataxia	Developing diabetes, hearing loss, progressively deteriorating functional status [60]	m.A3243G (tRNA leucine)
		Aphasia, subtle loss of muscle strength in right arm [61]	
		Migraine-like headaches, muscle damage [62]	
		disturbances of consciousness, ventilatory failure [63]	
		Aeizures, transient sensory disturbances, weakness, visual impairment, cognitive impairment [64]	m.G13513A (Complex1 ND5)
Maternally inherited diabetes and deafness (MIDD)	Chronic kidney disease, deafness, loss of oral sensation, neuropathy, hearing loss, myopathy	Proteinuria, glomerular abnormality progressive renal failuare [65]	m.A3243G (tRNA leucine)
		Hearing loss, loss of oral sensation, dysarthria, drolling [66]	
		Hearing loss, central nervous system diseases, myopathy, cardiac disease, nephropathy, underweight [10]	
Pearson syndrome	Failure to thrive, malabsorption	Refractory anemia, digestive system failuare, bone marrow failure, metabolic disorders, gastrointestinal symptoms, renal disorders, pancreatic exocrine insufficiency [11]	Large deletion of mtDNA
		Malabsorption, lactic acidemia, sideroblasts on bone marrow evaluation, microcephaly [67]	
Kearns–Sayre syndrome (KSS)	Heart block, growth retardation, external ophthalmoplegia, vestibular dysfunction	Vision loss, progressive external ophthalmoplegia, retinitis pigmentosa, heart block, vestibular dysfunction, growth retardation [68]	Large deletion of mtDNA
		Photophobia, nystagmus, sensorineural hearing loss, tremor, progressive cerebellar ataxia [69]	
Chronic progressive external ophthalmoplegia (CPEO)	Loosing control of eyelids and eye movements, ptosis	Central neurogenic hyperventilation, restriction of eye movement and ptosis [70]	Multiple deletion of mtDNA
		Low birth weight and congenital deafness [71]	
		Hemifacial weakness, dysarthria, mental retardation, sensorineural hearing loss [72]	mtDNA mutation with mitochondrial protein-encoding nuclear DNA mutations [73]
		Ophthalmoplegia, strabismus, loosing control of extraocular muscles [71]	
Leigh syndrome	Neurological symptoms, cardiac dysfunction, dyspraxia	Blepharoptosis, ptosis, inability to walk, sensory deficits, fever [74]	m.T9176C (MT-ATP6)
		Delay of psychomotor development, cardiomyopathy [75]	m.A14453G (Complex1 ND6)
		Bilateral exotropia, nystagmus bilateral, childhood-onset neuromuscular regression [76]	Nuclear *NDUFAF5* gene mutation, (Complex 1 NDUFAF5)
Barth syndrome	Heart failure	Left ventricular hypertrophy, heart failure with metabolic crisis [77]	Nuclear *TAZ* gene C640T (mitochondrial acyl chain composition remodeling enzyme)
Friedreich’s ataxia	Ataxia, gait unsteadiness, cardiomyopathy	Chest pain, dyspnea, palpitation, left ventricular hypertrophy [78]	Nuclear *FAZ* gene (mitochondrial iron metabolism related enzyme)
		Blindness, sensorineural deafness [79],	

**Table 2 cells-12-02494-t002:** Drugs for clinical trials for treating mitochondrial diseases.

Treatment	Mechanisms	Target Disease	Clinical Trial /NCT Number
Coenzyme Q10	Diffusible electron carrier of the mitochondrial respiratory chain [145], lipid peroxidation interfering antioxidants [146,147]	Mitochondrial diseases	Phase 3 (Completed) /NCT00432744
RAXONE (Idebenone)	Transfer electrons directly to complex III by bypassing malfunctional complex I [148,149,150,151]	LHON (Leber hereditary optic neuropathy)	Phase 4 (Completed) /NCT02774005
Vatiquinone (EPI-743, PTC-743)	Inhibiting 15-lipoxygenase (15-LO) [152] leads to increased GSH levels and decreased oxidized GSH [153,154]	Friedreich ataxia	Phase 3 (Active) /NCT05515536
		Mitochondrial disease with refractory epilepsy	Phase 2/3 (Active) /NCT04378075
		Mitochondrial respiratory chain diseases	Phase 2 (Active) /NCT01370447
Nicotinamide Riboside (NR)	Precursor of Nicotinamide adenine dinucleotide (NAD+); increasing intracellular NAD+ level increases mitochondrial function and mitochondrial number [155,156,157,158]	Mitochondrial myopathy disorder	Phase 2 (Active) /NCT05590468
KL1333	Interacts with NQO1 and acts as precursor of NAD+, recovers deficiency of mitochondrial respiratory chain [159]	Mitochondrial disease	Phase 2 (Active) /NCT05650229
Sonlicromanol (KH176)	ROS-redox modulator through interaction with the Thioredoxin System, which is a major antioxidant and redox signaling cellular system [160]	m.A3243G causing mitochondrial diseases	Phase 2 (Completed) /NCT04165239
Bocidelpar (ASP0367)	Agonist of PPARδ, enhances fatty acid oxidation, mitochondrial respiration and oxidative metabolism, which further leads to increment of skeletal muscle genes expression [161,162,163]	Primary mitochondrial myopathy	Phase 2/3 (Active) /NCT04641962
Mavodelpar (REN001, HPP593)	Agonist of PPARδ upregulates oxidative stress defense genes, contributes to attenuating oxidative stress [164]	Primary mitochondrial myopathy	Phase 2 (Active) /NCT04535609
Elamipretide (SBT-272)	Binding mitochondrial inner membrane cardiolipin; rescues dysmorphology of mitochondria [165,166]	Primary mitochondrial myopathy	Phase 3 (Active) /NCT05162768
		Barth syndrome	Phase 3 (Completed) /NCT03098797
		Mitochondrial dysfunction in age-related macular degeneration	Phase 2 (Completed) /NCT03891875
		Friedreich Ataxia	Phase 1/2 (Active) /NCT05168774
Skyclarys (Omaveloxolone)	Nrf2 degradation inhibitor; upregulates the expression of antioxidant gene, downregulates the expression of pro-inflammatory genes, and enhances mitochondrial biogenesis [167,168]	Friedreich’s ataxia	FDA approved
Deoxynucleosides Pyrimidines (Deoxycytidine dC and Deoxythymidine dT)	Enhances mtDNA maintenance by bypassing malfunctional mitochondrial pyrimidine salvage pathway [169]	Mitochondrial depletion syndromes with neurological phenotypes dysfunction	Phase 2 (Active) /NCT04802707

**Table 3 cells-12-02494-t003:** Non-chemical clinical treatments for mitochondrial diseases.

Treatment	Mechanisms	Target disease	Clinical trials
LUMEVOQ (GS010, rAAV2/2-ND4),	MT-ND4 deficiency rescue via allotopic expression of normal MT-ND4 using recombinant AAV [170,171,172]	LHON (Leber hereditary optic neuropathy)	Phase 3 (Active) /NCT03293524
NR082 (rAAV2-ND4),			Phase 2/3 (Active) /NCT03153293
ScAAV2-P1ND4v2	MT-ND4 deficiency rescue via allotopic expression of normal MT-ND4 using self-complementary AAV [173]		Phase 1 (Active) /NCT02161380
Mitochondrial augmentation	Replacement of dysfunctional mitochondria with healthy-exogenous donor mitochondria using in vitro uptake [174]	mtDNA depletion disease (Pearson syndrome)	Phase 1/2 (Active) /NCT03384420
Mesoangioblasts (MABs)	Intra-arterial injection of in vitro cultured patient-autologous mesoangioblasts, which harbor far fewer mtDNA mutations despite a much higher mutation load in patient [175,176]	Mitochondrial myopathy with m.A3243G mutation	Phase 2 (Active) /NCT05962333

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
