# Peer review of "Clinical Approaches for Mitochondrial Diseases"

_cells, 2023, doi:10.3390/cells12202494_

Round 1

Reviewer 1 Report

The manuscript is very well written and quite comprehensive. 

Two small items that could be fixed in the galley proof stage:

Line 130: sentence starting with "Since patients with LHON have high levels of mutated DNA heteroplasmy..."; recommend removing the word "Since".

Table 3: second column, bottom row, remove the word "their" from the sentence "...mtDNA mutations despite a much their higher mutation load in patient".

Author Response

First of all, we would like to give our appreciation for pointing out the parts we overlooked. Newly revised words in the main text are yellow shaded for clarity. Since new sentences have been added in response to the other reviewer's feedback, the line numbers have also changed. Location of adjustments in manuscript are indicated with “Line ###” in this response.
Revised manuscript file is attathced.

Revision #1: Line 130: sentence starting with "Since patients with LHON have high levels of mutated DNA heteroplasmy..."; recommend removing the word "Since".

Response: Thank you for your point out, we removed a word “Since” for clarity of sentence in “Line 141”.

Revision #2: Table 3: second column, bottom row, remove the word "their" from the sentence "...mtDNA mutations despite a much their higher mutation load in patient".

Response: I realized that I made such oversight, and I appreciate your help in pointing it out. Word “their” is removed in revised manuscript in “Table 3 2nd column, bottom row”.

Reviewer 2 Report

The manuscript entitled “Clinical Approaches for Mitochondrial Diseases” provides detailed information on mitochondrial diseases caused by mtDNA mutations, and the mechanisms of current treatments for mitochondrial diseases and clinical trials. This review focuses on treatments that are currently undergoing clinical trials for mitochondrial disease, with initial explanations of several mitochondrial diseases. The manuscript is well organized and scientifically sound. The manuscript needs minor revision before it can be accepted for publication.

1.     The authors need to add more recent references for all the mitochondrial diseases and their treatments mentioned in the review.

Author Response

First of all, thank you for your insightful feedback.
Our newly revised manuscript is attached below, responses are written in this box.

In the case of case study references, due to their rarity of mitochondrial diseases, it is possible that references before 2010 were used as case studies. In the manuscript, due to the inclusion of references that are akin to original papers in specific fields, or, in terms of paper for suggesting methods for diagnosis, there might be several references before 2010. We have rechecked to see if references before 2010 have been cited, even if they are not original papers or case studies, and I would appreciate it if you could take this into consideration.
In response to the reviewer's new suggestions, we have added new references with recent publications to enhance credibility of our manuscript. Added references with publication dates are described below.

- Reference no. [52] (doi : 10.1093/hmg/ddab156, Date : 2021)
For line 104-106,
“The manifestation of these symptoms is expected to depend on a certain threshold of heteroplasmy, which varies depending on the type of disease and the individual car-rying the mutated mtDNA”

- Reference no. [82] (doi : 10.3389/fopht.2022.1077395, Date : 2023)
For line 129-132
“Patients with LHON also have a higher likelihood of experiencing neurological ab-normalities, including peripheral neuropathy and myopathy, similar to other mito-chondrial diseases, compared to the general population”

- Reference no. [100, 101] (doi : 10.3390/jpm13010147, 10.1186/s12929-023-00966-8, Date of both : 2023)
For line 154-156
“One of the most common mutation among the patients with different types of mtDNA mutations is m.A8344G, which affects the mitochondrial tRNA lysine”

- Reference no. [101] also inserted into line 159-160, “Mutations in tRNA-coding genes induce a global impairment of mtDNA-encoded pro-teins rather than affecting certain complexes or pathways”

- Reference no. [135] (doi : 10.1111/joim.13047, Date : 2020)
For line 210-211
mtDNA deletion-related diseases appear to have less dependency on maternal inheritance”

- Reference no. [139] (doi : 10.1038/jhg.2011.97, Date : 2011)
For line 113-114
“The positions of mtDNA deletions are random, and the deletions are usually large, leading to overall mitochondrial dysfunction”

Reviewer 3 Report

      In this research, the authors reviewed the status of clinical approaches for mitochondrial diseases. In my opinion, the current version of this manuscript fits the scope of Cells and could be accepted after minor revision.

My specific comments are in detail listed below:

1.     In the discussion part, the authors may better point out the challenge of current strategies to treat mitochondrial diseases.   

2.     In the introduction, how mitochondria status affect the immune status of related mitochondrial diseases could be added since the role of ATP and AMPK in affecting immune status may be affected by mitochondria. Some references should be added to this part including 10.1002/adma.202206121.

3.     Some minor mistakes exist in this review. The authors should carefully check it.

4.     In the part “Clinical Approach with Treatments Involving Chemical Compounds’, the authors could predict the potent usage of some OXPHOS inhibitors to treat related mitochondrial diseases. Some references should be added to this part including 10.1002/advs.202207608.

5.     How DNA repair or damage affect the occurrence or development of mitochondrial diseases.

6.     Some references are out of date (before 2010). Some recent publications may be better.

Author Response

First of all, we would like to give our appreciation for your insightful feedback. In order to address the raised points, several sentences are added or changed. Newly revised sentences in the main text are yellow shaded for clarity and indicated with “Line ###” in this response.
Newly revised manuscript is attached below.

Revision #1. In the discussion part, the authors may better point out the challenge of current strategies to treat mitochondrial diseases.

Response: Thank you for your kind suggestion. In the manuscript, “Line 550-554” describes remaining challenges of treatments with broad intends, Excluding the inherent efficacy issues associated with each drug, I believe the most significant challenge in terms of mitochondrial disease therapeutics for drugs is that their indirect activities are ineffective for the main causes. So, I would like to cautiously inquire whether this might not be adequately explained by the existing sentence.
For non-chemical treatments, we concerned that point out for current challenges are required. We added few sentences for further description in “line 565-571” and displayed below in the box [].
[ Many of these types of non-chemical treatments have been developed, however, they still have limitations in that their approach is more targeted towards specific tissues with direct symptoms rather than the entire body's mitochondria. Mutated mitochondria are typically spread throughout the entire body in an individual, rather than being tissue-specific (PMID : 28933354, 203014036). Therefore, even if mitochondrial populations in one tissue are restored, it might lead to subsequent disorders in other tissues. Considering this, further research is required. ]

Revision #2. In the introduction, how mitochondria status affect the immune status of related mitochondrial diseases could be added since the role of ATP and AMPK in affecting immune status may be affected by mitochondria. Some references should be added to this part including 10.1002/adma.202206121.

Response: As suggested by the reviewer, adding a section of mitochondrial dysfunction with AMPK protein would be more beneficial for the manuscript.
Therefore, in the Introduction section, I have added an explanation of how mitochondrial dysfunction affects the AMPK protein and leads to various outcomes. I also included a description of the ongoing research in this regard in “line 63-68”
Furthermore, for Revision #4, I have included information about clinical approaches with this information. References generously provided by the reviewer have been greatly helpful in exploring this field. Both references (doi 10.1002/adma.202206121, 10.1002/advs.202207608) have been included in “line 66”.
added sentences are displayed below in the box [].
[
Mitochondrial defect also affects the closely associated protein, AMPK[26]. This, in turn, impacts the AMPK’s protein activity of regulating antitumor immunity [27,28]. In terms of this, research is being conducted to induce OXPHOS depression in mito-chondria, which affects AMPK and further promotes cancer immunotherapy[29,30]. Studies to control the population of dysfunctional mitochondria by utilizing involve-ment of AMPK protein in mitochondrial biogenesis are also in progress [31,32]. ]

Revision #3. Some minor mistakes exist in this review. The authors should carefully check it.

Response: I realized that I made such oversight, and I appreciate your help in pointing it out.
several changes are described below.
- word correction, echibit -> exhibit in “Line 191”
- Grammar correction, defects -> defect in “Line 280”
- Word “their” is removed in revised manuscript in “Table 3 2nd column, bottom row”
- removed blank in “Line 331”
- Adjust thickness of lines in Table1-3

Revision #4. In the part “Clinical Approach with Treatments Involving Chemical Compounds’, the authors could predict the potent usage of some OXPHOS inhibitors to treat related mitochondrial diseases. Some references should be added to this part including 10.1002/advs.202207608.

Response: I acknowledge the importance of research on using OXPHOS inhibitors to regulate mitochondrial function in tumors and explore treatment possibilities, particularly in the context of mitochondrial-related disorders. However, as the 'Clinical Approach with Treatments Involving Chemical Compounds' section of the manuscript primarily discusses approaches aimed at restoring mitochondrial dysfunction caused by mitochondrial gene mutations, we decided to include this explanation in the Introduction along with my response to revision #2, as it complements the main text more effectively (included in the response for revision #2).

Revision #5. How DNA repair or damage affect the occurrence or development of mitochondrial diseases

Response: The DNA repair mechanisms of mitochondria remain a less extensively researched field, thus in-depth references regarding their relevance to diseases with recent studies are difficult to find. The occurrence of mtDNA deletions due to malfunctioning DNA-related mechanisms could be regarded as a case of DNA damage (10.1038/ng.f.94, 10.1016/j.tig.2019.01.001) Since all mitochondrially-encoded proteins play important roles in mitochondria, mutations in coding genes within mitochondria can directly lead to overall mitochondrial dysfunction. Also described in “Line 36-38”

Revision #6. Some references are out of date (before 2010). Some recent publications may be better.

Response: In the case of case study references, due to their rarity of mitochondrial diseases, it is possible that references before 2010 were used as case studies. In the manuscript, due to the inclusion of references that are akin to original papers in specific fields, or, in terms of paper for suggesting methods for diagnosis, there might be several references before 2010. We have rechecked to see if references before 2010 have been cited, even if they are not original papers or case studies, and I would appreciate it if you could take this into consideration.
However, we have added new references with recent publications to enhance credibility of our manuscript. Added references with publication dates are described below.

- Reference no. [52] (doi : 10.1093/hmg/ddab156, Date : 2021)
For line 104-106,
“The manifestation of these symptoms is expected to depend on a certain threshold of heteroplasmy, which varies depending on the type of disease and the individual car-rying the mutated mtDNA”

- Reference no. [82] (doi : 10.3389/fopht.2022.1077395, Date : 2023)
For line 129-132
“Patients with LHON also have a higher likelihood of experiencing neurological ab-normalities, including peripheral neuropathy and myopathy, similar to other mito-chondrial diseases, compared to the general population”

- Reference no. [100, 101] (doi : 10.3390/jpm13010147, 10.1186/s12929-023-00966-8, Date of both : 2023)
For line 154-156
“One of the most common mutation among the patients with different types of mtDNA mutations is m.A8344G, which affects the mitochondrial tRNA lysine”

- Reference no. [101] also inserted into line 159-160, “Mutations in tRNA-coding genes induce a global impairment of mtDNA-encoded pro-teins rather than affecting certain complexes or pathways”

- Reference no. [135] (doi : 10.1111/joim.13047, Date : 2020)
For line 210-211
mtDNA deletion-related diseases appear to have less dependency on maternal inheritance”

- Reference no. [139] (doi : 10.1038/jhg.2011.97, Date : 2011)
For line 113-114
“The positions of mtDNA deletions are random, and the deletions are usually large, leading to overall mitochondrial dysfunction”

Reviewer 4 Report

Dear Authors, please find some suggestions that could be useful to further improve your manuscript.

- Please insert a brief section in which you describe the mitochondrial compartment. This will help s non expert in the field to understand the effects of the mtDNA mutations to the mitochondria.

- Please also insert a section in which you describe the mitochondrial quality control system (fusion and fission, mitophagy and mitobiogenesis). Recently, they have been demosnstrated to have great therapeutic  values in several mitochondrial diseases, such as in PMID: 26206091, 35858578, 33665835 , 32061767, 26341273

-Please provide a reference regarding this sentences: "For medical treatments of LHON patients, a phase 4 clinical trial has been completed, and Raxone (Idebenone) has been approved by the European Medicines Agency (EMA) for use with LHON patients. However, it has not received approval from the FDA."

- Figure 2 is a little bit confusing. I encourage to improve it to make the main concepts more clear.

- Table 3 refers to "non-drug clinical treatment". However, the caption 5 there is wrtitten "" with non-chemical treatments". Please uniform. 

Author Response

First of all, we would like to give our appreciation for your insightful feedback. In order to address the raised points, several sentences are added or changed. Newly revised sentences in the main text are yellow shaded for clarity and indicated with “Line ###” in this response.
our newly revised manuscript is attached below,

Revision #1. Please insert a brief section in which you describe the mitochondrial compartment. This will helps non expert in the field to understand the effects of the mtDNA mutations to the mitochondria.

Response: Thank you for your advice that has helped to enhance the content of our manuscript. In the section describing diseases related to specific compartment malfunctions, such as LHON and Leigh syndrome, which are being well-researched, in addition to diseases involving overall mitochondrial dysfunction rather than specific compartment disorders, I have included some content that corresponds to the reviewer's suggestions, along with additional revisions as provided below in this response.

The description of Complex 1 and the ND1-6 proteins in section of LHON has been added to the existing manuscript in “Line 137-139” and described below with box []
[ND1–6 are the components of the NADH-ubiquinone oxidoreductase complex (complex 1), and therefore, mutations associated with LHON cause defects in complex 1, which has the function of catalyzing reactions in the electron transport chain]

The description of MT-ATP6 in the section on Leigh syndrome in “Line 199-201” described below with box []
[The defect in MT-ATP6 leads to the dysfunction of mitochondrial complex 5, which converts ADP to ATP, eventually resulting in a lack of energy.]

Revision #2 Please also insert a section in which you describe the mitochondrial quality control system (fusion and fission, mitophagy and mitobiogenesis). Recently, they have been demosnstrated to have great therapeutic  values in several mitochondrial diseases, such as in PMID: 26206091, 35858578, 33665835 , 32061767, 26341273

Response:
I have included information about clinical approaches with this information in introduction. References generously provided by the reviewer have been greatly helpful in exploring this field.
descriptions are added in “line 68-71” and displayed below in box []
[In the context of mitochondria-related autophagic processes, which involve the regu-lation of dysfunctional mitochondria populations, this approach holds the potential for treating mitochondrial diseases and is actively being investigated[33-37].]

Revision #3 Please provide a reference regarding this sentences: "For medical treatments of LHON patients, a phase 4 clinical trial has been completed, and Raxone (Idebenone) has been approved by the European Medicines Agency (EMA) for use with LHON patients. However, it has not received approval from the FDA."

Response: Thank you for your point outs. I have added the completed NCT number for the Phase 4 clinical trial (NCT02774005). I have included the EMA approval product number for RAXONE (Idebenone) in the text (EMEA/H/C/003834) in “Line 283-285”.

The article about FDA denial in Santhera’s homepage has been removed. Former address :
(https://www.santhera.com/docs/default-source/2016/2016-07-14_pr-regulatory-update_e_final.pdf?sfvrsn=4)
But there are some articles with the context of FDA denial of RAXONE.
(https://www.mda.org/press-releases/fda-turns-down-santhera-pharmaceuticals%E2%80%99-request-accelerated-approval-idebenone-treat)

Revision #4 Figure 2 is a little bit confusing. I encourage to improve it to make the main concepts more clear.

Response: In line with the reviewer's comments, we also found that it is confusing due to the extensive content (from both Table2,3.). In our intend of presenting both chemical and non-chemical drugs in a single figure, it unavoidably leads to an increase in text and a more complex illustration. To facilitate a more concise explanation, we made several modifications.
1. We removed the boxes and replaced them with drug names (Bold, underlined) along with a more simplified description of features (Colored with light gray).
2. We simplified the figure to clarify the indications of each drug’s acting point.
we hope that our changed version of figure 2. meets our intention to introduce table 2, 3. comprehensively.

Revision #5
Table 3 refers to "non-drug clinical treatment". However, the caption 5 there is written "" with non-chemical treatments". Please uniform.

Response: Thank you for your acknowledgement, we revised “non-drug clinical treatment” to “non-chemical clinical treatments” in table 3. Line 391

Round 2

Reviewer 4 Report

Dear authors,

I appreciated you considered all my suggestions.

I think the manuscript has been improved and it is ready for publication.

Best regards